# Does age matter?—Efficiency of mechanical food break down in *Tupaia belangeri* at different ages

Achim H. Schwermann[1]*, Julia A. Schultz[2], Eberhard Fuchs[3], Andreas Leha[4], Jürgen Hummel[5], Ottmar Kullmer[6,7], Patrick Steuer[8], Thomas Martin[2]

1 LWL-Museum of Natural History, Westphalian State Museum and Planetarium, Münster, Germany,
2 Department of Palaeontology, Institute of Geosciences, Universität Bonn, Bonn, Germany, 3 Deutsches Primatenzentrum, Göttingen, Germany, 4 Department of Medical Statistics, University Medical Center Göttingen, Göttingen, Germany, 5 Department of Animal Sciences, Ruminant Nutrition, University of Göttingen, Göttingen, Germany, 6 Department of Palaeoanthropology, Senckenberg Research Institute and Natural History Museum Frankfurt, Frankfurt am Main, Germany, 7 Department of Paleobiology and Environment, Institute of Ecology, Evolution, and Diversity, Goethe University, Frankfurt, Germany, 8 Senzyme GmbH, Troisdorf, Germany

* achim.schwermann@lwl.org

**Data Availability Statement:** The statistical data of the analysis are included in the manuscript. The basic number series (chitin particles per sample,

## Abstract

The relationship of food comminution and individual age in *Tupaia belangeri* is investigated. It is hypothesized that with increasing age the performance of the molar dentition decreases due to progressive tooth wear. While this relationship is well-documented for herbivores, age-related test series are largely lacking for insectivorous mammals. 15 individuals of *Tupaia belangeri* were fed exclusively with mealworms, and their faeces were analyzed for the number and size of chitin particles. The exoskeleton of a mealworm is resistant to digestive fluids in the gastrointestinal tract, and the size of individual chitin particles indicates the effectiveness of mechanical comminution that occurs in the oral cavity during mastication. It is hypothesized that a more precise occlusion of the dentition results in smaller particle size. Although individuals of all ages (juvenile, adult, and senile) were able to effectively process mealworms with their dentition prior to digestion, a larger area of very large chitin particles (98% quantile of all particles in senile animals as compared to in the same quantile in adults) in the feces of senile animals was detected. Even though the particle size of indigestible material is irrelevant for the digestive process, these findings either document somatic senescence in the functionality of the teeth, or alternatively a change in chewing behaviour with age.

## Introduction

Among many other characteristics, the ability to process food intensively with a complex dentition is a unique feature of mammals [1]. Mammalian teeth show distinctive adaptations to a certain type of food which is broken down mechanically inside the oral cavity prior to chemical processing [e.g. 2,3]. In many cases, crushing into smaller pieces is what makes swallowing

size of particles) is published online. DOI: http://doi.org/10.25625/6PGJBW.

**Funding:** Funding for this study was provided by Deutsche Forschungsgemeinschaft (DFG) to TM (MA 1643/17).

**Competing interests:** The authors have declared that no competing interests exist.

possible in the first place and increases the surface area of the food for better accessibility to biochemical digestion [4–6].

With some exceptions, mammals usually have two generations of teeth, a derived condition from the basal tetrapod bauplan, in which continuous tooth replacement for the entire lifetime is found [7]. Due to the fact that mammals have evolved a permanent dentition in adults, they develop a precise occlusion between antagonistic teeth with complementary contacts. This adaptation is associated with the inevitable consequence of an age-related increasing dental wear [e.g. 8–10] leading to a reduced effectivity of mastication in senile individuals. Various adaptations to maintain long-term tooth functionality have been observed in many, mainly herbivorous, mammalian groups, such as hypsodonty and duplication of crown structures [e.g. 7,11].

Chewing is the repetition of occlusal movements of the lower jaw to mechanically prepare a food bolus. The process of chewing is accompanied by physical wear. Over the course of a life-time, abrasion and attrition lead to a permanent loss of enamel and the exposure of dentin [12–14]. In particular protruding structures in the antagonists (e.g., cusps and crests) are affected by wear, but also areas of the crown without antagonistic dental structures (e.g., stylar cusps, basins, cingula) show specific wear patterns [15]. Antagonistic dental contacts cause light reflecting wear facets, which are restricted mostly to the occlusal areas of the tooth [5,16,17]. Attritional wear is usually attributed to tooth-tooth contacts, and abrasion to tooth-food-tooth contacts [18]. Tooth wear in mammals does not only have destructive effects due to the loss of dental hard tissue. In many mammal species with selenodont cheek teeth, for example, wear is necessary to functionally adapt the antagonists through sharpening cutting edges [19]. In many carnivore and rodent mammals, moderate wear of the crown is also necessary for the formation of sharp cutting blades and for the dentition to become fully functional (e.g., sharpening of carnassials in Carnivora and chisel-like incisors in Rodentia) [20–23].

In many herbivorous mammals, functional wear and thus the removal of the dental tissue is required for the formation of characteristic shearing surfaces (secondary functional crown shape after [24]), often combined with ever-growing molars to compensate for the loss of dental tissue [25]. This way, the functionality of an occlusal surface can increase by wear or at least kept stable for a certain period of time in the life of an individual. The breakdown of a steep relief is correlated with a decrease of enamel edges and thus shear-cutting structures. In this respect, insectivorous mammals are equipped with very steep pointed cusps on their relatively low-crowned molars. Wear facets occur along the steep flanks of cusps on the primary enamel coat. Progressing wear removes the enamel coat and levels the crown with increasingly age [15,26,27]. Thus, teeth of insectivorous mammals with a high crown relief (with pointed cusps) are possibly losing their effectivity through wear during progressing ontogeny.

Our study aims to investigate the influence of increasing age on the efficiency of mastication in insectivorous mammals. We tested the hypothesis that young individuals perform a more efficient dental break-down of food then older ones, using individuals of different age of the tree shrew *Tupaia belangeri* (Scandentia). The genus *Tupaia* Raffles, 1821 [28] includes 15 species and numerous subspecies, present mainly in Southeast Asia [29]. Generally, the genus is characterized as insectivorous, but some species consume also a large percentage of fruit within their diet [30,31]. *T. belangeri* (Wagner, 1841 [32]) is ecologically similar to the slightly smaller *T. glis* (Diard and Duvaucel, 1820 [33]). The latter is known to feed mainly on insects and other arthropods, as well as sweet and oily fruits [34,35]. Langham [34] also reports about preying on young birds and other small vertebrates as part of the diet, but this was not confirmed by analysis of stomach contents and faeces. *T. belangeri* may reach a body mass of 270 g [34,36]. Thus, from wildlife observations and additional morphological analysis [30], a general insectivorous lifestyle is assumed here for *T. belangeri*. Because of its high crown relief, *Tupaia*

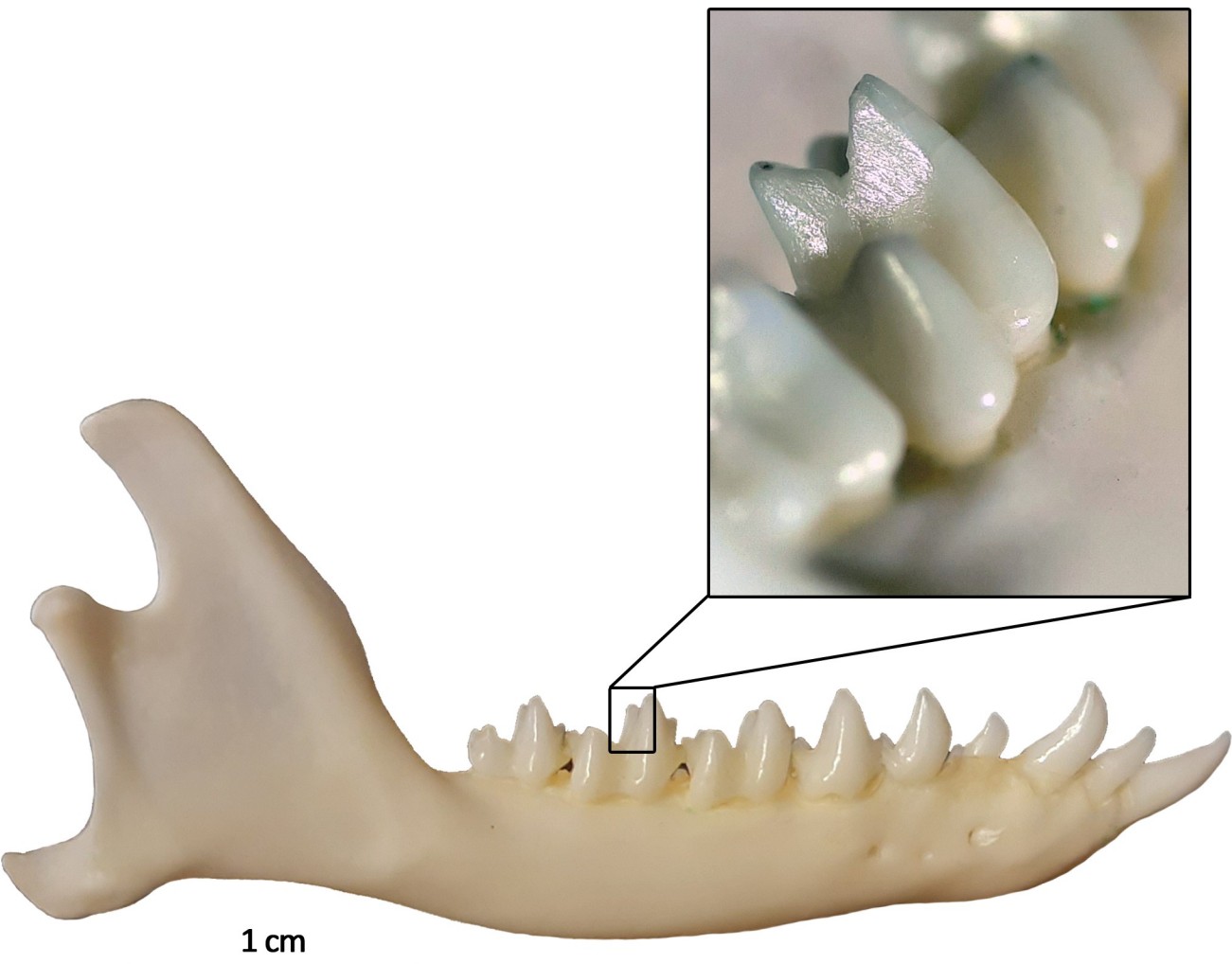

**Fig 1. Right dentary of *Tupaia glis lacernata* (Institute for Geoscience Paleontology MaÜ 70).** The reflecting facets at the steep flanks of the molars are resulting from tooth-to-tooth wear).

*belangeri* dentition appears to be best suited to insectivory whereas other tree shrew taxa exhibit certain adaptations in the teeth towards frugivory (Fig 1) [30].

## Material and methods

### Feeding experiments

Tree shrews (*Tupaia belangeri*) from the breeding colony of the German Primate Center (Göttingen, Germany) were used for the feeding experiments. The animals were housed singly from puberty onwards in steel cages (50×80×125 cm) under a 12-hour-light/12- hour-dark cycle (lights on at 8:00 am) with 60±7% relative humidity and an ambient temperature of 27 ±1˚C (for details, see [36]). Housing under routine conditions was not changed for the present study. Three juvenile (one male/two females; mean age four months), six adult (three males/ three females; mean age four years) and six senile animals (one male/five females; mean age seven years) were investigated. The life span of tree shrews under laboratory conditions is approximately ten years and thus 3–4 times longer than the life span of mice and rats [37].

All animal housing and handling was in accordance with the European Communities Council Directive of September 22, 2010 (2010/63/EU) for the Care and Use of Laboratory Animals. Tree shrews of the genus *Tupaia* are predominantly insectivorous and are used to receive meal worms as food in the author's laboratory (EF). Animals were used to daily handling; they were not suffering and were not exposed to stressful situations. Thus, ethical approval was not required for this study as stated by the Animal Welfare Committee and the Animal Welfare Officer of the German Primate Center (documented under no. E3-21). Following this research, the animals remained undisturbed in the animal facility.

The animals were fed larvae of mealworms (*Tenebrio molitor*). As shown in other experimental series, those larvae can be characterized as a soft insect diet [38–40]. The tree shrews were fed exclusively with the larvae for three-day minimum prior to collecting feces, to ensure that the gastrointestinal system was free from other food items. Access to the food and water was without limitation to the individuals (*ad libitum*). After the initial three days, the animals were fed larvae continuously for five days. Individual feces samples (between 2 and 5 per animal) were collected under dimmed light before the lights were turned on after a slight massage of the hypogastrium. The feces samples of the 15 *Tupaia belangeri* (for details see Table 1) were collected under attendance of one of the authors (EF). Samples were collected separately by individual and calendar day. Samples were transferred directly into plastic vials, stored at -20˚C and shipped on dry ice to the University of Bonn for analysis.

The samples were resolved in hydrogen peroxide-solution (5%) to separate chitin particles from each other. All wet samples were washed with distilled water and filtered through filter paper (retention 12–15 μm); retained material was then spread on filter papers. Subsequently, the wet samples were freeze-dried to prevent the chitin particles from sticking to the filter paper. The dried samples were sorted manually from clumped particles other than chitin (e.g., hay, hairs, litter) under the microscope.

Further preparation of samples followed the protocols of Nørgaard et al. [41–44]. To capture form and size of the chitin particles of each sample, the particles were distributed evenly over the glass plate of a flatbed scanner (Canonscan 9000F Mark II). Particles were manually shifted, in order to have no overlapping particles. Each scan was performed with a resolution of 2400 dpi (Fig 2A).

An automatic analysis of the images to capture form and size of particles by the scanner was not possible, because very thin chitin particles are translucent and thus could not be captured. Therefore, each image was processed manually, masking every particle using the Adobe Photoshop CS5 software package. For each critical item, it was individually decided whether it was

**Table 1. Descriptive statistics.** Data are summarized grouped by age class.

| parameter | | juvenile | adult | senile |
|---|---|---|---|---|
| number of animals | | 3 | 6 | 6 |
| number of feces samples | | 14 | 14 | 23 |
| number of particles | | 194567 | 114910 | 72422 |
| feces sample size [number of particles] | | | | |
| | mean ± sd | 13898 ± 11357 | 8208 ± 8045 | 3149 ± 5220 |
| | median (min; max) | 11657 (1188; 49382) | 5534 (1687; 30669) | 1393 (298; 25937) |
| particle size [mm] | | | | |
| | mean ± sd | 0.63 ± 1.6 | 0.62 ± 1.7 | 1.2 ± 3.7 |
| | median (min; max) | 0.088 (5e-04; 61) | 0.075 (0.00055; 48) | 0.075 (0.00051; 89) |

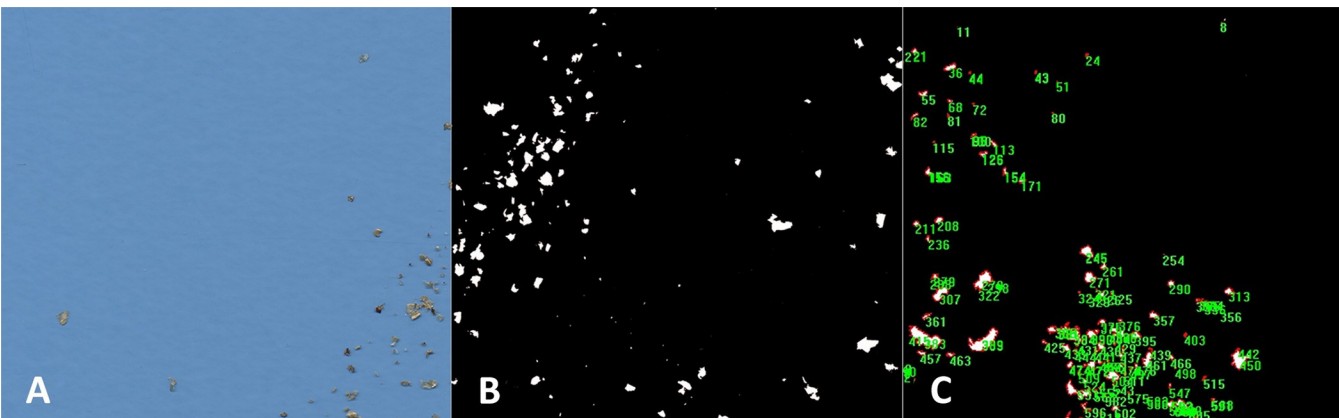

**Fig 2. Digital analysis of a feces sample.** (A) scan image of chitin particles spread at the glass plate of the flatbed scanner. (B) abstracted chitin particles in white; all foreign particles are removed. (C) processing and analysis by Image-Pro Analyzer, measuring greatest length, width, and surface area.

chitin or color information of the background. All particles other than chitin left in the sample were excluded digitally. Dust particle and fabrics were removed by scaling down all masked particles about three pixels and increasing about the same amount afterwards. Based on the resulting mask, the background color then was changed to black, and the particles' color to white (Fig 2B).

The analysis of the produced image was done by Image-Pro Analyzer 6.3 (Media Cybernetics, Rockville, USA). Each particle per sample was automatically numbered (Fig 2C). Greatest length and width (vertically to each other), as well as surface area of each particle were measured. For the computed analysis of the particles, 0.0005 mm$^2$ was chosen as minimum area size and 0.025 mm as minimum width. Particles smaller than the minimum values were excluded. The measurements were exported to MS excel.

The data collected are published here http://doi.org/10.25625/6PGJBW.

## Statistical analyses

Several feces samples were collected per animal and the size of the particles in each feces sample was measured. Number of particles per feces sample as well as particle sizes (in mm$^2$) were summarized as mean+-sd and median (minimum; maximum). Taking the single particle as experimental unit the influence of the age class on the log particle size was assessed with mixed effect models controlling for the number of particles per feces sample (sample size) and the interaction of that number with age class, and including animal and sample as random factors to control for their associated intra-class correlation (as random intercept models) while also accounting for their nested structure (particles per sample per animal) leading to this model (in R-code for easy comprehension):.lmer(log(particlesize) ~ ageclass*samplesize + (1|animal/sample)). Similarly, the effect of age class on number of particles per sample was modeled using mixed effect models. Additionally, the particle sizes were binarised twice (extremely large as $> = 98\%$ quantile and extremely small as $< = 2\%$ quantile) and logistic mixed effect regression models were fit to the binary outcomes.

P values for the regression coefficients were obtained using Wald-statistics approximation (treating t as Wald z). In the logistic regression models, Satterthwaite's degrees of freedom method was used.

The significance level was set to $\alpha = 5\%$ for all statistical tests. All analyses were performed with the statistics software R (version 3.6.1, [45]) using in particular the R-package lme4

(version 1.1.21 [46]), the mixed effect linear/logistic regression models and sjPlot (version 2.6.3 [47]) as well as lmerTest (version 3.1.0 [48]) to obtain p values of the regression coefficients.

## Results

Data from 51 feces samples of 15 animals were collected and analysed (Table 1 and Fig 3).

The number of particles per sample was significantly smaller in the senile age class as compared to the juvenile class (estimate: -10728, p < 0.01) and in tendency also for adults compared to juveniles (estimate: -5658, p = 0.08, Fig 4A, S1 Table).

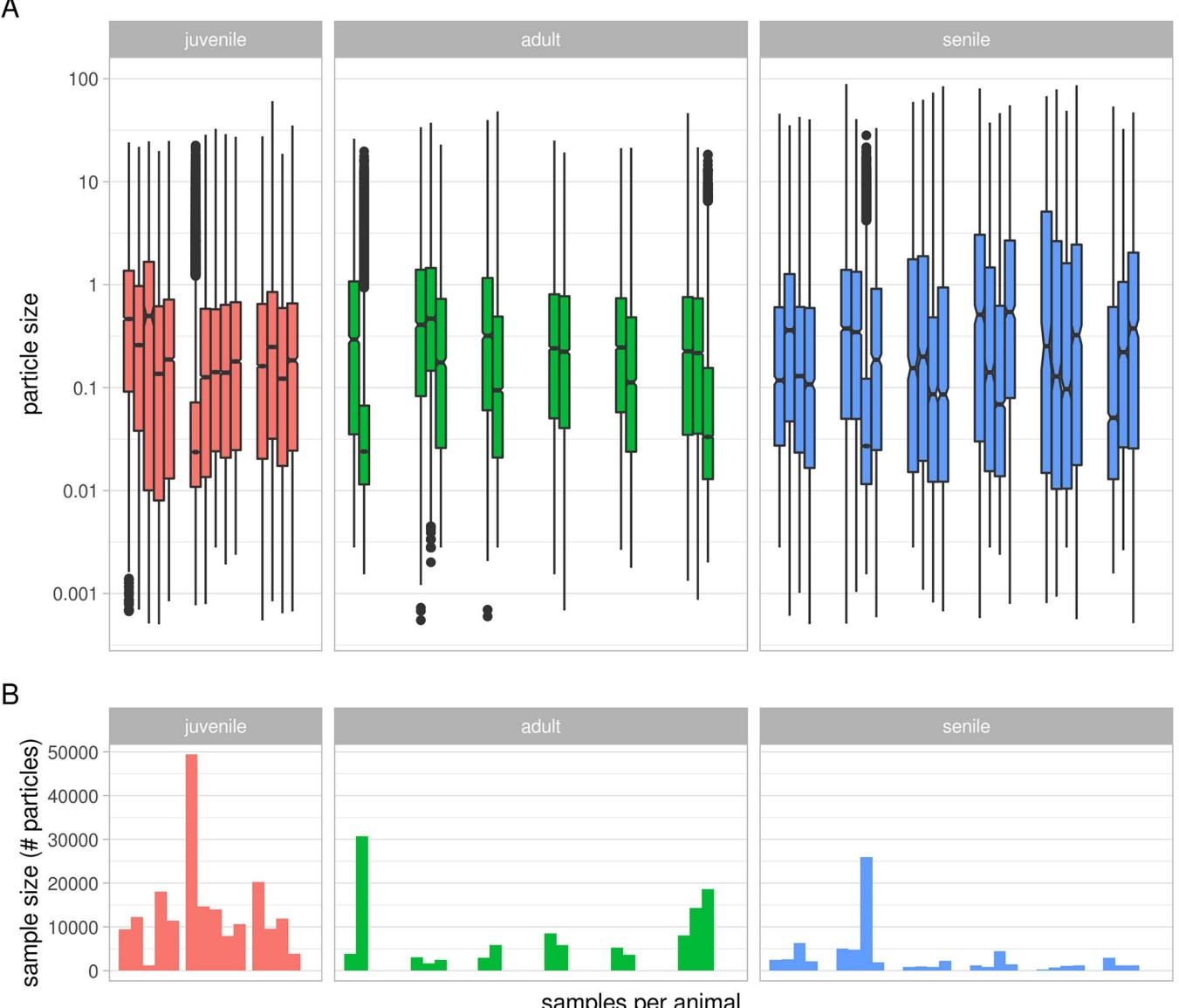

**Fig 3.** Number of particles (B) and particle sizes (A) per sample. Samples (x axis) are grouped by animal and colored by age class (red/left: Juvenile, green/middle: Adult, blue/right: Senile). The distribution of the number of particles (A) is shown on logarithmic scale.

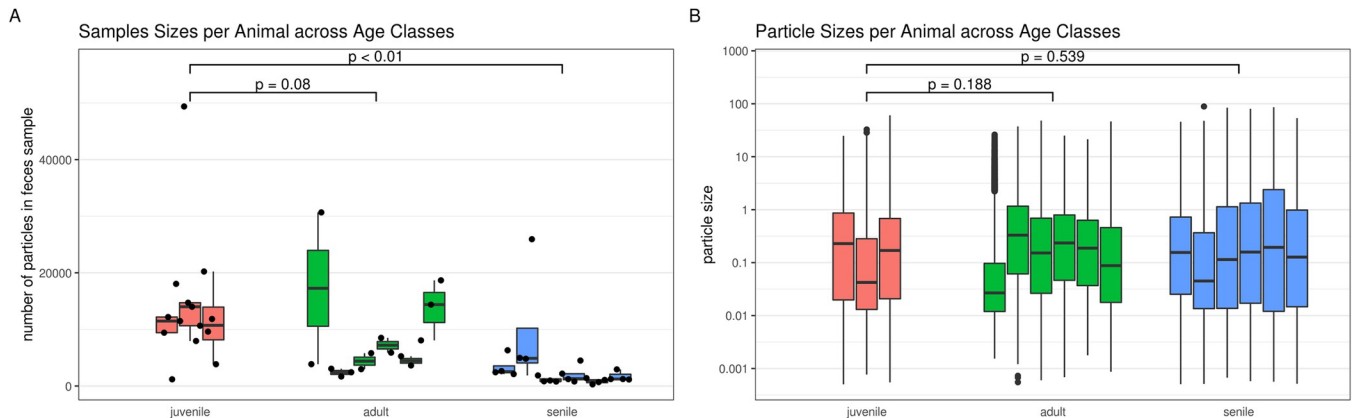

**Fig 4.** Number of particles per sample (A) and particle sizes (B) compared between age classes. P values are from linear mixed effect models accounting for nested random effects (samples under animals). The particle sizes in the model are controlled for number of particles per sample and the interaction of this number and age class and the main effects for age class are reported.

The mean of particle sizes (on log scale) was not significantly different between the age classes (Fig 4B, S2 Table). But the odds for particles to be extremely small (2% quantile, smaller than 0.0028 mm$^2$) were significantly larger in the juvenile samples compared to adult (p = 0.017) and the senile classes (p < 0.001, Fig 5A, S3 Table). In the opposite direction, the odds for a particle to be extremely large (98% quantile, larger than 6.8 mm$^2$) were significantly lower in the juvenile samples as compared to samples from the adult (p = 0.038) or the senile (p < 0.001) animals (Fig 5B, S4 Table).

## Discussion

The results of this study show a change in the distribution of particle size with age. This concerns the increase in size of the particles of the 98% quantile. It most likely has little relevance to digestive efficiency in relation to the diet used. While this could be conveniently interpreted as an indication of a loss of dental functionality with age (assuming a similar number of chews

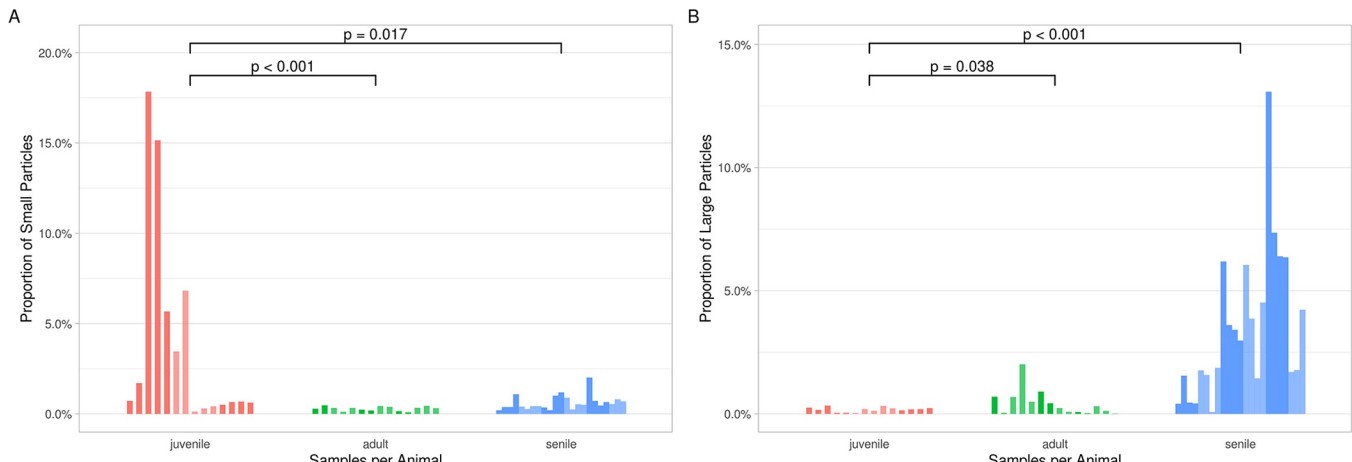

**Fig 5.** Proportions of large (A) and small (B) particles in each sample. Samples are grouped by animal with different shades to visually differentiate between the two. P values are from logistic mixed effect models accounting for nested random effects (samples under animals) and controlling for number of particles per sample and the interaction of this number and age class. Reported are the main effects for age class.

per amount of food consumed), we cannot rule out the possibility that this is due to a reduction in chews per amount of food consumed in older, more experienced animals.

Four main findings are evident in this study, which support the following discussion: 1) the tribosphenic molar pattern of *Tupaia belangeri* allows effective breakdown of food already at a juvenile stage of ontogenetic development, when the teeth are used during mastication; 2) no significant change is detected in the size of individual chitin particles from juvenile to adult *T. belangeri* in general when feeding exclusively on mealworms (significant differences are limited to the content of extremely small or large particles; see 4) for this); 3) the functionality of the molars is maintained until senile age in regard to breaking down mealworms; 4) even though the average particle sizes do not differ between age groups, we observed that juvenile animals produce significantly more extremely small ($< 0.0028$ mm$^2$) and senile animals significantly more extremely large particles ($> 6.8$ mm$^2$).

The tribosphenic molar morphology of *Tupaia* shows pointed main cusps and shearing edges with steep wear facets. Compared to other insectivorous species, *Tupaia* therefore appears to be well equipped with teeth suitable for effective shear-cutting and crushing of insect body parts. This is also confirmed by Selig et al. [30], who quantified morphological adaptations to insectivory within Scandentia, especially in *T. belangeri*. The experiments show that *Tupaia* of all age classes are able to effectively break down the mealworms offered. As expected, the biomechanical equipment in a tribosphenic dentition makes the exploitation of insects as a food source, or presumably arthropods in general, perfectly possible. The shearing facets and crests developed as structures in the tribosphenic molar, evolved as an adaptation to insectivory, and occur universally in many extant groups in which generalized insectivory is retained (i.e., Ameridelphia, Dasyuromorpha, Afrosoricida, Macroscelidea, Eulipotyphla, Chiroptera, Dermoptera, and Scandentia) [49].

The following discussion is based on the general assumption that the comminution of insectivorous exoskeletons is purely mechanical by the dentition. It was observed that some insectivorous taxa (e.g., the tenrecs *Echinops* and *Hemicentetis*) regurgitate complete or partial earthworms to chew and swallow them a second time [Francke 1961, cited in 50–52]). This rumination-like behavior—although probably pathological or atypical—observed in some insectivores is very likely to affect particle size of the swallowed food items. The same is very likely for coprophagy, which is known in some non-herbivore taxa [e.g. 53–55]. However, during the time of feeding experiments, none of these behaviors were observed in *Tupaia*, so the effects of mastication a second time will not be discussed here.

*Tupaia*-individuals of all age groups chewed the mealworms in such a way that chitin fragments of the exoskeleton could be detected in the faeces, each representing only small parts of the original surface. Moore and Sanson [39] studied the dissolubility of nitrogen of mealworm larvae in trypsin (digestive tract enzyme). The study concludes that a larva with intact exoskeleton is less affected by trypsin, while the breakdown of the larva in two pieces increased the effect significantly. Similar observations were made by Prinz et al. [38], who punctured mealworms and crickets to increase the effect of simulated "digestion" by hydrochloric acid. Earlier, Kay and Sheine [56] fed prepared chitin particles (two different sizes) to a small strepsirrhine primate (*Galago senegalensis*) and came to the conclusion that smaller particles are more affected by digestion than bigger ones. They generally were able to show that the smaller a section of a mealworm is, the more it is affected by digestion. But they also showed that pushing the dental break-down to its limits does not increase the absorption of nutritive substances. If a single mealworm is cut into a high number of equal pieces (cutting into 16 pieces or more), the digestibility does not increase further. Those studies underlined the necessity of opening the exoskeleton of mealworm larvae for an effective digestion within the intestines of insectivorous mammals. In this context, Jablonski and Crompton [57] observed that individuals of the

haplorrhine primate *Tarsius bancanus* execute 7–14 chewing cycles to break down a cricket before swallowing it. They noticed that these few chewing cycles were not sufficient to homogenize the bolus. Rather, it can be assumed that the multiple openings of the exoskeleton of the cricket alone are sufficient for swallowing and prepare it for digestion in the gastrointestinal tract. The generalized hypothesis that smaller food particles are correlated with better digestibility seems applicable to herbivorous species, but, as shown by Kay and Sheine [56], it is less important for insectivorous mammals. The mealworm feeding experiments show that the proportion of extremely large particles, which may reduce the digestibility of the complete larva, is more meaningful than that of extremely small particles when interpreting the efficiency of the breakdown of mealworms. In the present study, juvenile and adult *Tupaia belangeri* show an average particle size of less than one square millimeter ($0.63 \pm 1.6$ mm$^2$ in juveniles and $0.62 \pm 1.7$ mm$^2$ in adults), whereas the value in senile animals is $1.2 \pm 3.7$ mm$^2$. Those values represent only a small percentage of the outer surface of the exoskeleton of a mealworm (several 100 mm$^2$) and suggest a good digestibility for the non-chitin material in all age classes.

A differentiation in the chewing behaviour of the three age groups is found, which in turn has an effect on efficiency and thus consequently on the individual energy balance (compare 4) at the beginning of the discussion). Juvenile animals tend to mechanically homogenize mealworms to a much greater extent than adult and senile animals do, which can be seen when looking at the extreme values only (extremely small and extremely large particles). They generally produced a very high proportion of extremely small particles, and simultaneously the number of chitin particles per sample is significantly higher than in senile individuals. On the contrary, the proportion of particles larger than 6.8 mm$^2$ (98% quantile) is larger in the senile animals than in the younger age groups. This corresponds to the slightly, non-significant higher average particle size of senile individuals. The significantly higher proportion of extremely large chitin particles in the feces of the senile animals only partially confirms the working hypothesis (decreasing efficiency of food processing with individual age). At first glance, it can be concluded that the efficiency declines with age, fitting the assumptions that with older age the degree of tooth wear increases and the precision of occlusion decreases. But, the number of chewing cycles has also a direct effect on the resulting particle size, as shown by Santana et al. [2]. In their study, they fed bats with scarabaeid beetles, counted the chewing cycles and investigated the number and size of chitin particles in the resulting feces. For insectivorous taxa, excessive mechanical comminution of food is not necessary. In terms of efficiency, the energy gain from the food must be compared to the energy input of the previous mechanical comminution work [compare 38,39,56]. With increasing comminution of food by mastication, a threshold value is unavoidably passed at which the increasing energy input meets a stagnating energy gain. An excessive mechanical comminution of the food is therefore, from a certain point on, no hindrance to digestive efficiency, but not efficient in terms of energy balance.

The most important observation of the series of experiments is that no significant differences in mealworm comminution occur in the different age classes of *Tupaia* in general. The significant differences are limited to the proportion of very small, or very large particles, while the size of the particles in the three age classes is not generally significantly different. This contradicts the initial hypothesis that younger individuals break down their food more efficiently than old individuals do. While in this study the effectiveness of mastication is assumed for all age groups, the question of efficiency remains unanswered. We can show that for the understanding of the latter, other parameters must be considered. The number of chewing strokes per amount of food, respectively the energy input, is crucial. This is included in the following working hypothesis: All individuals of different age groups perform a similar number of chewing strokes for the mastication of a certain amount of food. The chewing stroke of juvenile and

adult animals therefore would have to be considered more effective as it left fewer large particles behind. This would mean that the age has a direct influence on the particle size, and be an indication of loss of optimal functionality–most likely due to wear. However, at least for the food used in the present study, this decrease in effectiveness cannot be considered a constraint in terms of energy or nutrient acquisition. The amount of putative tooth wear would not be sufficient to be a cause of general restriction of the organism. Ideally, inspections of teeth of animals from the different age groups in future studies could yield additional indication for changes in dental morphology reducing the teeth's efficiency.

Alternatively, senile animals might perform fewer chewing strokes, which also would result in more large particles. The increased proportion of large particles, which was observed in this study, would then be correlated with the number of chewing strokes, and less with the abrasion of the teeth. This would mean that senile animals chew more efficiently in terms of energy balance.

The results presented here are preliminary and point out the necessity of further feeding experiments to address the question whether the functionality of molars change with age in insectivorous mammals to the full extent. On the other hand, they indicate direct approaches for these further investigations of the feeding behaviour and help to address the questions of effectivity and efficiency, especially in the tribosphenic molar pattern.

## Supporting information

**S1 Table. Model coefficients from a linear mixed effect model for the sample size by age class.**
(DOCX)

**S2 Table. Model coefficients from a linear mixed effect model for the particle size by age class controlled for number of particles per sample in the interaction.**
(DOCX)

**S3 Table. Model coefficients from a mixed effect logistic regression model for extremely large particle size by age class controlled for number of particles per sample in the interaction.**
(DOCX)

**S4 Table. Model coefficients from a mixed effect logistic regression model for extremely small particle size by age class controlled for number of particles per sample in the interaction.**
(DOCX)

## Acknowledgments

We would like to thank all members of the Research Unit FOR 771 for their discussion and support. We thank Georg Oleschinski (Institute of Geosciences, University of Bonn) for photography. We are grateful to Daniel Jansen for his patient processing of the samples.

## Author Contributions

**Conceptualization:** Achim H. Schwermann, Julia A. Schultz, Jürgen Hummel, Ottmar Kullmer, Thomas Martin.

**Data curation:** Achim H. Schwermann, Eberhard Fuchs, Jürgen Hummel, Patrick Steuer.

**Formal analysis:** Andreas Leha.

**Funding acquisition:** Ottmar Kullmer, Thomas Martin.

**Investigation:** Achim H. Schwermann, Eberhard Fuchs, Andreas Leha, Patrick Steuer.

**Methodology:** Achim H. Schwermann, Jürgen Hummel.

**Project administration:** Achim H. Schwermann.

**Supervision:** Julia A. Schultz.

**Writing – original draft:** Achim H. Schwermann, Julia A. Schultz, Eberhard Fuchs, Andreas Leha, Thomas Martin.

**Writing – review & editing:** Jürgen Hummel, Ottmar Kullmer, Patrick Steuer.

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
