## [Editor Report · Decision Letter 0]

24 Aug 2021

PONE-D-21-23819

Does age matter? - Efficiency of mechanical food break down in Tupaia belangeri at
different ages

Dr. Achim Hermann Schwermann

Dear Dr. Schwermann,

I hope this message finds you well. I am writing in regards to the recent withdrawal
of your PLOS ONE manuscript, PONE-D-21-23819 "Does age matter? - Efficiency of
mechanical food break down in Tupaia belangeri at different ages." I apologize for
any confusion the withdrawal may have caused you. Your extension request had been
missed by our staff, therefore your manuscript was mistakenly withdrawn.

Your manuscript has now been reinstated and returned to the 'Submissions Needing
Revision' folder in your Editorial Manager account. Please resubmit your manuscript
files when your are ready.

Your resubmission is due on Oct 08 2021 11:59PM.

Please do not hesitate to contact us at plosone@plos.org if you have any queries. Thank you, and have a great
day.

With best wishes, 

Arianna Casabonne

Staff

PLOS ONE

---

## [Author Response · Author response to Decision Letter 0]

13 Sep 2021

The passage concerning animal welfare has been added to the Material and Methods
chapter: "All animal housing and handling was in accordance with the European
Communities Council Directive of September 22, 2010 (2010/63/EU) for the Care and
Use of Laboratory Animals. Tree shrews of the genus Tupaia are predominantly
insectivorous and are used to receive meal worms as food in the author’s laboratory
(EF). Animals were used to daily handling; they were not suffering and were not
exposed to stressful situations. Thus, ethical approval was not required for this
study as stated by the Animal Welfare Committee and the Animal Welfare Officer of
the German Primate Center (documented under no. E3-21). Following this research, the
animals remained undisturbed in the animal facility."

---

## [Decision Letter · Decision Letter 1]

2 Jun 2022

PONE-D-21-23819R1Does age matter? -
Efficiency of mechanical food break down in Tupaia belangeri at different
agesPLOS ONE

Dear Dr. Schwermann,

Thank you for submitting your manuscript to PLOS ONE. After careful consideration, we
feel that it has merit but does not fully meet PLOS ONE’s publication criteria as it
currently stands. Therefore, we invite you to submit a revised version of the
manuscript that addresses the points raised during the review process.

In this revised version, please provide your original data and more information or
justification about your methods (use of 'individual' as a random factor, age
categories used instead of exact age, description of mean particle size per sample).
You will also find several language corrections in the attached file.

Please submit your revised manuscript by Jul 17 2022 11:59PM. If you will need more
time than this to complete your revisions, please reply to this message or contact
the journal office at plosone@plos.org. When
you're ready to submit your revision, log on to https://www.editorialmanager.com/pone/ and select the 'Submissions
Needing Revision' folder to locate your manuscript file.

Please include the following items when submitting your revised
manuscript:A rebuttal letter that responds to each point raised by the academic
editor and reviewer(s). You should upload this letter as a separate file
labeled 'Response to Reviewers'.A marked-up copy of your manuscript that highlights changes made to the
original version. You should upload this as a separate file labeled
'Revised Manuscript with Track Changes'.An unmarked version of your revised paper without tracked changes. You
should upload this as a separate file labeled 'Manuscript'.If you would like to make changes to your financial disclosure,
please include your updated statement in your cover letter. Guidelines for
resubmitting your figure files are available below the reviewer comments at the end
of this letter.

We look forward to receiving your revised manuscript.

Kind regards,

Cyril Charles

Academic Editor

PLOS ONE

Journal Requirements:

Reviewers' comments:

Reviewer's Responses to Questions

**Comments to the Author**

1. If the authors have adequately addressed your comments raised in a previous round
of review and you feel that this manuscript is now acceptable for publication, you
may indicate that here to bypass the “Comments to the Author” section, enter your
conflict of interest statement in the “Confidential to Editor” section, and submit
your "Accept" recommendation.

Reviewer #1: (No Response)

2. Is the manuscript technically sound, and do the data
support the conclusions?

Reviewer #1: Yes

3. Has the statistical analysis been performed
appropriately and rigorously? 

Reviewer #1: No

4. Have the authors made all data underlying the
findings in their manuscript fully available?

Reviewer #1: No

5. Is the manuscript presented in an intelligible
fashion and written in standard English?

Reviewer #1: Yes

6. Review Comments to the Author

Reviewer #1: PONE-D-21-23819R1

reviewed by Marcus Clauss, Zurich

This manuscript describes the particle sizes in faeces of Tupaia of different ages
fed with mealworms. The results show that younger animals have more and smaller, and
senile animals less and larger particles in their faeces. While this is an
indication of an effect of tooth wear, the authors correctly discuss that it could
be an effect of different chewing behaviour. The approach is sound and the results
are - for me - exciting.

The original data is not given, which is not acceptable nowadays in my view.

There are some language issues with the text (e.g., the German "probe" is often used
instead of "sample"), and in particular consistency across references is not good
(e.g., supplementary tables have different labels in the main text compared to the
attached supplements).

Tables - both in the main text and in the excel supplement - are not given in
standard layout quality - authors, please take care to provide properly layouted
tables with legends in the next version.

A lot of the wording in the methods and discussion remains unclear. E.g., in the
methods, it is mentioned thhat 'individual' is used as a random factor in the mixed
models, but in order for thhat to make sense, one should mention that several faecal
samples were taken per individual. I made a lot of comments in that respect in the
attached word file.

The discussion often contradicts the results - see the comments - this must be
corrected. Also, there are some redundancies in the discussion.

In terms of methods, my main question is - these are breeding-colony-animals, so the
eact age should be known - whhy was age not investigated as a covariable rather than
making "categories". This can be justified - e.g. if the effect of age was clearly
not linear - but at least plots where x=age and y=proportion of very small and very
large particles should be given. If these plots show non-linear trends that are
difficult to analyse, then making age categories is particularly justified.

Fig. 1 should be replaced by a photo of a Tupaia tooth.

The description of mean particle size per sample is lacking (just the mean of all
counted prticles?) - and Table 1 should provide both, length in mm and area in mm2
for the mean size for a sample.

Please see the attachhed word file for details.

sincerely marcus clauss

7. PLOS authors have the option to publish the peer
review history of their article (what does this mean?). If published, this will
include your full peer review and any attached files.

If you choose “no”, your identity will remain anonymous but your review may still be
made public.

**Do you want your identity to be public for this peer review?** For
information about this choice, including consent withdrawal, please see our
Privacy Policy.

Reviewer #1: **Yes: **Marcus Clauss

---

## [Author Response · Author response to Decision Letter 1]

26 Aug 2022

Dear Plos One Team,

we have followed or implemented all comments, corrections and suggestions of the
reviewer. Please see the coverletter "Response to Reviewers".

We would like to thank Markus Clauss for his support.

Best regards, on behalf of the author team,

Achim Schwermann

---

## [Editor Report · Decision Letter 2]

30 Aug 2022

Does age matter? - Efficiency of mechanical food break down in Tupaia belangeri at
different ages

PONE-D-21-23819R2

Dear Dr. Schwermann,

We’re pleased to inform you that your manuscript has been judged scientifically
suitable for publication and will be formally accepted for publication once it meets
all outstanding technical requirements.

Kind regards,

Cyril Charles

Academic Editor

PLOS ONE
---

## [Editor Report · Acceptance letter]

1 Sep 2022

PONE-D-21-23819R2 

Does age matter? - Efficiency of mechanical food break down in *Tupaia
belangeri* at different ages 

Dear Dr. Schwermann:

I'm pleased to inform you that your manuscript has been deemed suitable for
publication in PLOS ONE. Congratulations! Your manuscript is now with our production
department. 

Kind regards, 

on behalf of

Dr. Cyril Charles 

Academic Editor

PLOS ONE